# Reactivity of shape-controlled crystals and metadynamics simulations locate the weak spots of alumina in water

R. Réocreux[1,6], É. Girel [2,3], P. Clabaut[1], A. Tuel[3], M. Besson [3], A. Chaumonnot[2], A. Cabiac[2], P. Sautet [4,5] & C. Michel [1]

The kinetic stability of any material in water relies on the presence of surface weak spots responsible for chemical weathering by hydrolysis. Being able to identify the atomistic nature of these sites and the first steps of transformation is therefore critical to master the decomposition processes. This is the challenge that we tackle here: combining experimental and modeling studies we investigate the stability of alumina in water. Exploring the reactivity of shape-controlled crystals, we identify experimentally a specific facet as the location of the weak spots. Using biased *ab initio* molecular dynamics, we recognize this weak spot as a surface exposed tetra-coordinated Al atom and further provide a detailed mechanism of the first steps of hydrolysis. This understanding is of great importance to heterogeneous catalysis where alumina is a major support. Furthermore, it paves the way to atomistic understanding of interfacial reactions, at the crossroad of a variety of fields of research.

[1] Univ Lyon, Ens de Lyon, CNRS UMR 5182, Université Claude Bernard Lyon 1, Laboratoire de Chimie, 69342 Lyon, France. [2] Direction Catalyse et Séparation, IFP Energies nouvelles, Rond-point de l'échangeur de Solaize, BP 3, 69360 Solaize, France. [3] Institut de Recherches sur la Catalyse et l'Environnement de Lyon, IRCELYON, CNRS UMR 5256–Univ. Lyon 1, 2, avenue Albert Einstein, 69626 Villeurbanne, France. [4] Department of Chemical and Biomolecular Engineering, University of California Los Angeles, Los Angeles, CA 90095, USA. [5] Department of Chemistry and Biochemistry, University of California Los Angeles, Los Angeles, CA 90095, USA. [6] Present address: Thomas Young Centre and Department of Chemical Engineering, University College London, Roberts Building, Torrington Place, London WC1E 7JE, UK. Correspondence and requests for materials should be addressed to P.S. (email: sautet@ucla.edu) or to C.M. (email: carine.michel@ens-lyon.fr)

The kinetic stability of solids in water is governed by their reactivity at the interface. Being able to understand and control their surface stability and desired properties is therefore at the heart of a variety of research fields: kinetics of drug release[1], corrosion of metals and alloys[2], lithium batteries[3], geochemistry[4] with in particular the re-equilibration of solid phases in presence of a liquid[5], water treatment[6], heterogeneous catalysis[7,8], from the preparation[9,10], the utilization to the degradation of the catalyst[11,12], etc. Several parameters have been identified as key in all these fields: the solid phase (which polymorph)[13], the nature of the surface exposed (kink, rugosity)[4], and the species in solutions (additives, ions, pH)[14–16].

For instance, alumina is an oxide that is used in catalysis as a support for metallic nanoparticles[17]. Its γ-allotrope (γ-Al$_2$O$_3$) in particular has remarkable properties as a support, long proven in the development of gas phase transformations[18–21]. However, water has detrimental effects on γ-Al$_2$O$_3$, either as a liquid[12] or even as steam[22]. While its polymorph α-Al$_2$O$_3$ is stable, γ-Al$_2$O$_3$ indeed transforms into various sorts of bulk hydroxides (AlO$_x$H$_y$) thereby seriously damaging the catalyst (decrease of surface area, sintering and encapsulation of catalytic particles)[23–25]. Albeit not fully described at the atomic scale yet, the decomposition of γ-Al$_2$O$_3$ was shown to proceed, at the macroscopic scale, through a sequential mechanism involving first dissolution of Al atoms, and then precipitation thereof into the undesired hydroxides[16,24]. These structural changes can be retarded by tuning the surface chemical composition: silica deposition[26], presence of metallic nanoparticles[12], or impregnation with metal ions[27]. They are also impacted by the content of the solution: pH[24] but also presence of polyols[28] or polyphenols[25]. The mechanism of action of these additives is believed to lie in their chemisorption on the surface that would make γ-Al$_2$O$_3$ water-resistant[28,29]. However little is known on which exposed facet(s) and what site(s) need to be particularly targeted for protection. The optimization of the structure of inhibitors is therefore a challenging task since the exact mechanism of the decomposition of γ-Al$_2$O$_3$ in liquid water remains unknown. Whilst the sub-nanometric description of γ-Al$_2$O$_3$ is difficult to reach experimentally, molecular simulations have allowed for the development of insightful atomistic models for amorphous[30] and nano-crystalline[31] γ-Al$_2$O$_3$. The model proposed by Digne et al.[31], which was obtained from density functional theory (DFT) calculations, has proven to depict properly the reactivity of the surface under various realistic gas phase conditions[19]. It is only recently that Ngouana-Wakou et al.[32] and Réocreux et al.[33] have explored the interaction of γ-Al$_2$O$_3$ with liquid water, performing ab initio molecular dynamics (AIMD) simulations. Although these simulations give insights on the relative affinity of water for the different facets and the structuration and dynamics of the liquid at the interface, they cannot capture the details of the decomposition mechanism. Al-O bond formation and scissions are indeed rare events on the time scale achievable today with AIMD. To overcome this limitation and force the system to react along a chosen reaction coordinate, biased AIMD simulations are required. Ab initio metadynamics is an example of such methods[34], and has been commonly used to model the reactivity between molecules in homogeneous media[35,36] or between isolated molecules and a solid surface[37,38]. Although common using classical force field for crystal growth and dissolution of molecular and ionic crystals[39,40], metadynamics has never been used to describe a reactive interface involving bonds with a partial covalent character (typically here Al–O and O–H bonds) hence requiring an ab initio description.

Here, we combine experiments and theory to provide the atomistic mechanism for the early-stage decomposition of γ-Al$_2$O$_3$ in liquid water. We experimentally demonstrate that the decomposition is initiated at the (110) facet by exposing four samples of various shapes of nanoparticles to a hydrothermal treatment in the presence of aqueous solution of inhibitors (xylitol and sorbitol) with varying concentrations. Performing ab initio metadynamics simulations, we probe the reactivity of the (110)/water interface and identify specific aluminum tetrahedral centers that are particularly reactive with water. We show that interfacial water molecules are involved in the mechanism, both as reactants for the hydration of aluminum and as intermediates for the proton reshuffling required by the decomposition mechanism. We show that the substitution of chemisorbed water molecules on the surface with xylitol locally renders the surface more hydrophobic and pushes water molecules away from the water-sensitive Al sites.

## Results

**Identification of the facet whence the decomposition initiates.** γ-Al$_2$O$_3$ particles mainly expose three facets commonly referred to with their set of Miller indices: the predominant and hydrophilic (110) facet, the less hydrophilic (100) facet, and the (111) facet (see Fig. 1a). In order to identify the facet(s) involved in the decomposition of γ-Al$_2$O$_3$, we have prepared four samples containing particles of different shapes (see Fig. 1g–j). These four samples therefore show four distinct proportions of facet areas, which have been quantified using X-ray diffraction (XRD) (see Supplementary Note 1). We have had them undergo hydrothermal treatment (i.e., 200 °C, 2 h, 14 bar autogeneous pressure) using various aqueous solutions of sorbitol (represented in Fig. 1b), a known inhibitor, and xylitol (represented in Fig. 1c), a shorter sugar polyol. We have observed a very strong alumina stabilization in the presence of both sorbitol or xylitol, suggesting that the minimal –(CHOH)$_5$– sequence, common to those two polyols, is essential to protect alumina from water. This is consistent with the work by Ravenelle et al.[28] showing that sorbitol is a good inhibitor while glycerol (C3-polyol) is not as efficient. To go beyond this qualitative statement, we have determined for each sample the minimum surface coverage of polyol at which the decomposition of alumina gets inhibited, that is no hydroxides can be detected by XRD after hydrothermal treatment (see Supplementary Note 2). This quantity, referred to as *inhibiting coverage*, is reached for concentrations of 4 g L$^{-1}$ of polyols for 2 g of alumina in 50 mL of solution and varies from 0.15 to 0.30 nm$^{-2}$ for xylitol, and from 0.20 to 0.35 nm$^{-2}$ for sorbitol, depending on the shape of the nanoparticles (see Supplementary Table 4). This strong variation of *the inhibiting coverage* with the distribution of facet surface areas (up to a factor of two in the case of xylitol) indicates that each facet does not interact equally with the two polyols.

In the case of a specific adsorption on one facet only, the so-determined *inhibiting coverage* increases linearly with the fractional surface area of this facet with a zero-intercept (see Supplementary Note 4). As shown in Fig. 1d–f, this is only with the fractional surface area of the hydrophilic (110) facet that the *inhibiting coverage* of both sorbitol and xylitol correlates with a zero-intercept function. We cannot establish such correlations with the other fractional surface areas. Following the same kind of reasoning, we show in Supplementary Note 5 that edges and kinks are unlikely to be involved as major active sites for hydrolysis. Therefore, the adsorption of sorbitol or xylitol is specific to the (110) facet and allows for the total inhibition of the decomposition of γ-Al$_2$O$_3$ in liquid water at similar partial coverage of 0.36 and 0.44 nm$^{-2}$ of (110) facet for xylitol and sorbitol, respectively (slopes on Fig. 1d). As a corollary, we can infer that the (110) facet must exhibit sites responsible for the hydrolysis of γ-Al$_2$O$_3$ and that the reactivity of these sites with water can be limited or even suppressed upon the adsorption of sorbitol or xylitol.

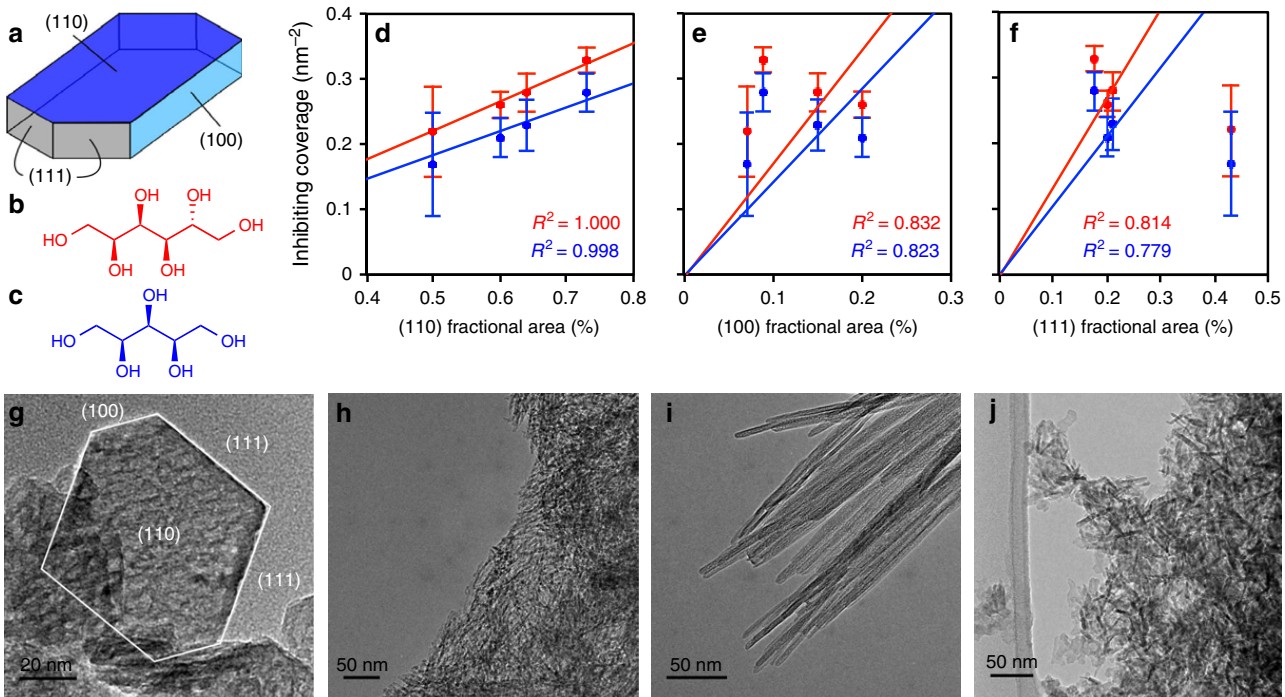

**Fig. 1** Implication of the (110) facet in the decomposition of γ-Al$_2$O$_3$. **a** General topology of a γ-Al$_2$O$_3$ nanoparticle exhibiting three major facets: (110) in dark blue, (100) in light blue and (111) in gray. **b, c** Structures of two inhibitors: **b**, sorbitol (red) and **c**, xylitol (blue). **d, e, f** Comparisons between the *inhibiting coverage* (sorbitol in red, xylitol in blue) and the fraction of exposed (110) (**d**), (111) (**e**), and (100) (**f**) surface areas. The experimental data sets are fitted to the zero-intercept linear model derived in Supplementary Note 4. The fitted curves are represented as straight lines. A correlation could only be established in the case of the (110) surface (**d**) with slopes of 0.367 ± 0.009 and 0.443 ± 0.005 nm$^{-2}$ for xylitol and sorbitol, respectively ($R^2 > 0.99$). Error bars are calculated from high-performance liquid chromatography (HPLC) analysis standard deviation (see Supplementary Note 3). **g, h, i, j** Transmission electronic microscopy images of the four differently shaped γ-Al$_2$O$_3$ nanoparticles ordered in increasing (110) fractional area: plates (**g**), fibers (**h**), rods (**i**), and commercial γ-Al$_2$O$_3$ (**j**)

**Identification of the atomic sites on the (110) facet whence the decomposition initiates**. To gain atomistic insight and identify the sites involved in the decomposition of γ-Al$_2$O$_3$ on the (110) facet, we have performed ab initio simulations using our recently developed model for γ-Al$_2$O$_3$/water interfaces[33]. The primitive cell of the model of the γ-Al$_2$O$_3$(110) surface shows four different surface Al sites: two octahedral and two tetrahedral sites, referred to as Al$_{(1)}$ and Al$_{(2)}$ for the former and Al$_α$ and Al$_β$ for the latter (see Fig. 2a, b). This surface is fully hydrated with five chemisorbed water fragments, the oxygen atoms of which are represented in blue in Fig. 2a, b. Some of these water molecules are dissociated, generating hydroxyl surface groups. Repeating the surface unit cell in the x and y directions, the resulting p(2 × 2) slab is surmounted by a 20 Å thick layer of liquid water that is not represented here for clarity. The iso-electric point of alumina being close to 8, we consider pure neutral water.[41,42] Since the decomposition of γ-Al$_2$O$_3$ transforms tetrahedral Al sites into octahedral Al centers, we have focused on the tetrahedral sites, Al$_α$ and Al$_β$, and their reactivity with water (see Fig. 2a). Both of them have a coordination number to alumina oxygen atoms (CN$_a$) of three and a coordination number to water oxygen atoms (CN$_w$) of one. Performing ab initio metadynamics, we have made each tetrahedral Al center specifically react along these two variables (see details in Methods). The statistical analysis of the variations of CN$_w$ and CN$_a$ allows for the construction of free energy landscapes (like the one given in Fig. 2c), the local depth of which assesses for the stability of the intermediates, transition states and products encountered along the course of the simulation (see details in Supplementary Note 6). Obtaining this mechanistic information is a key achievement in the quest of a

better understanding of reactive interfaces and kinetic stability of solids in water. In the case of Al$_α$, the free energy landscape mainly shows one deep minimum at (CN$_a$,CN$_w$) = (3,1), that is the initial structure (see Supplementary Fig. 6). Even within a span of 200 kJ mol$^{-1}$, no transition state to escape that deep well towards other minima has been identified. This suggests that Al$_α$ is very unlikely to react with liquid water. Conversely, the free energy landscape for Al$_β$ shows, over a span of 180 kJ mol$^{-1}$, a variety of minima spread over the nodes of a well-defined checkered pattern and associated with a decreased number of bonds to alumina surface oxygen atoms and increased number with water oxygens (see Fig. 2c). Al$_β$ therefore progressively detaches from the oxygen network of γ-Al$_2$O$_3$ and is hydrated with the surrounding water molecules. A more detailed look at the free energy surface indicates that this hydration is sequential and follows an addition/elimination mechanism. It is only when Al$_β$ has reached a total coordination number of 5 that an Al$_β$–O–Al bond cleavage occurs. Noticeably, five-coordinated species appear to be key intermediates in the decomposition of alumina as in the water-induced de-alumination of zeolites[43]. This holds true until (CN$_a$,CN$_w$) = (1,4) where Al$_β$ readily guests a water molecule to achieve an octahedral structure with (CN$_a$, CN$_w$) = (1,5). This last structure obtained from the simulation is represented in Fig. 2e and shows how Al$_β$ has been extracted from its initial position. Among the water molecules in the first coordination sphere, three of them were initially present on the surface as chemisorbed water molecules and the two others were physisorbed water molecules. Proton transfers are not included in the general reaction coordinates we designed. Nevertheless, the inspection of the trajectory shows that these early-stage steps of

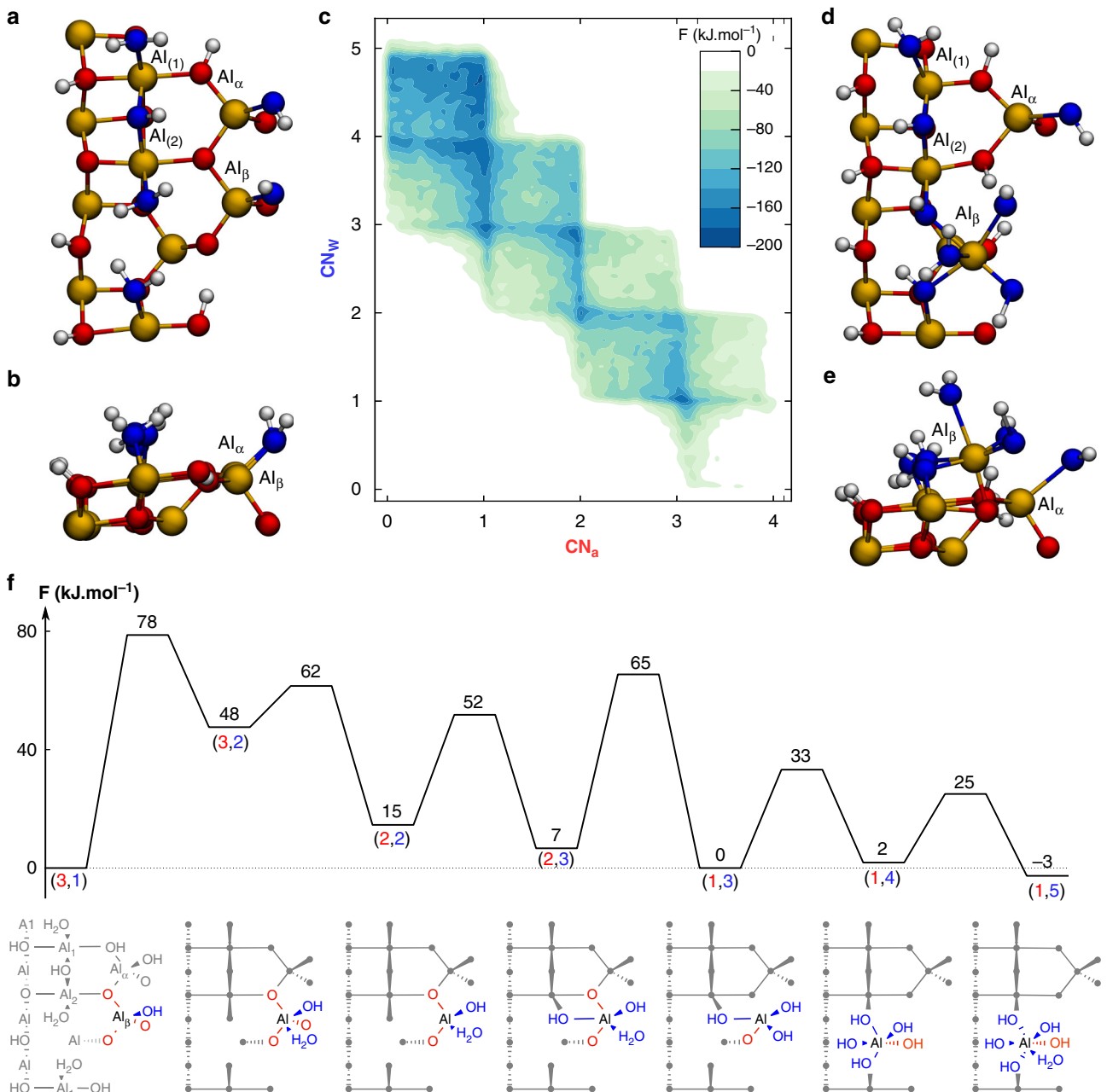

**Fig. 2** Exploration of the reactivity of tetrahedral Al$_\beta$ on γ-Al$_2$O$_3$ (110) from ab initio metadynamics. **a, b** Structure of the initial surface with chemisorbed water molecules or fragments (the oxygen atoms of which are shown in blue) adsorbed on tetrahedral Al$_\alpha$ and Al$_\beta$ and octahedral Al$_{(1)}$ and Al$_{(2)}$. **c** Free energy surface obtained from the metadynamics simulation on Al$_\beta$ using the coordination numbers to alumina oxygen atoms (CN$_a$) and to water molecules and fragments (CN$_w$) as variables. The simulation starts from the point of coordinates (CN$_a$,CN$_w$) = (3,1). **d, e** Structure of the last intermediate obtained from the hydration of tetrahedral Al$_\beta$ with (CN$_a$, CN$_w$) = (1,5). **f** Projected free energy profile with the structure of each intermediate and the corresponding (CN$_a$,CN$_w$). Yellow and white balls represent aluminum and hydrogen atoms, respectively. The color red is used for alumina oxygen atoms and the associated CN$_a$. The color blue is used for water oxygen atoms and the associated CN$_w$

the hydration are accompanied with the deprotonation of some of the coordinated water molecules and a widespread reshuffling of surface protons over about 1 nm (compare Fig. 2a, d). The amphoteric character of the hydrated material allows γ-Al$_2$O$_3$ to offer the proper local protonation level for the intermediates involved in its own decomposition into AlO$_x$H$_y$. In this respect, the trajectory reveals that physisorbed water molecules are involved, via a Grotthus mechanism, in this redistribution of surface protons. Physisorbed water molecules do therefore not only react but also help accommodating the protonation state of the surface to guest the final octahedral intermediate. This is

particularly striking for the water fragments bound to the octahedral Al$_{(1)}$ and Al$_{(2)}$, which need to be deprotonated to coordinate the fleeing tetrahedral Al$_\beta$. This study highlights the key role of water as a reactant and as a solvent in the stability of an oxide and pre-figures the role of the pH in the interface transformation. Ab initio metadynamics can hence provide a complete atomistic picture of the alteration mechanism of an iono-covalent solid in contact with a reactive solvent.

**Simulation with chemisorbed xylitol.** We have performed the same simulation substituting the water fragments bound to the

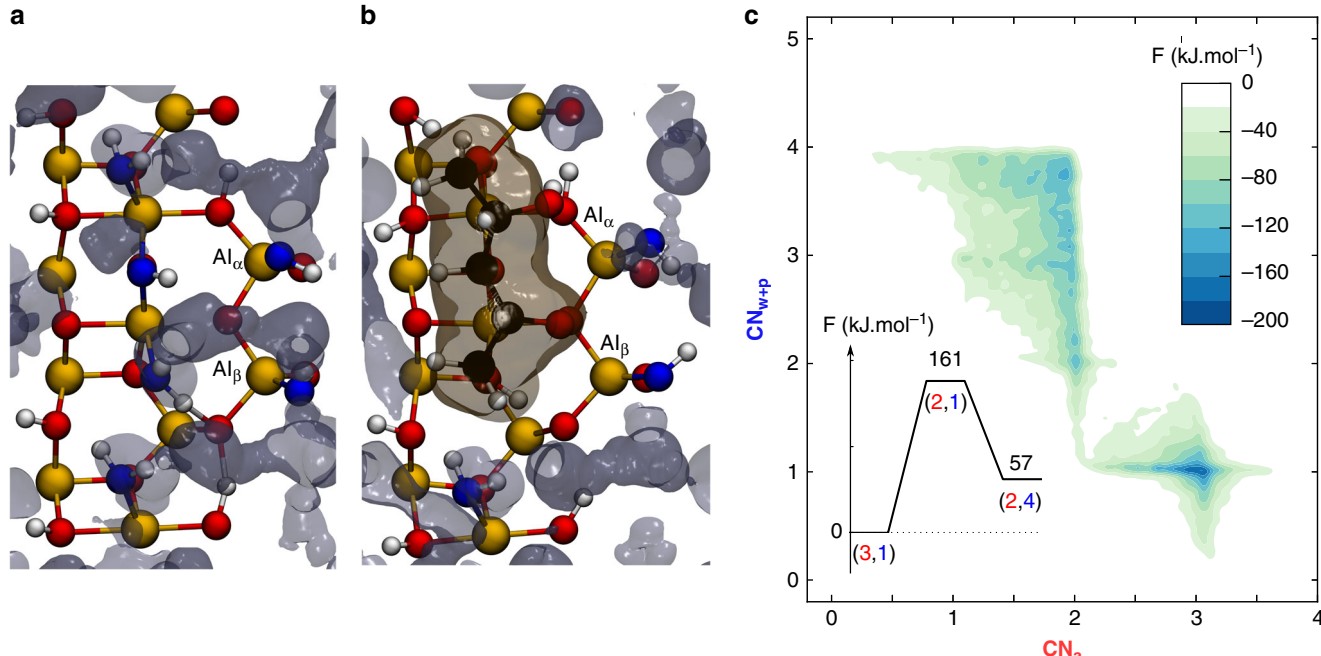

**Fig. 3** Inhibition of the decomposition of γ-Al₂O₃ in the presence of chemisorbed xylitol. **a** γ-Al₂O₃(110)/water interface with the average occupation volume of water in gray. **b** Geometry of adsorbed xylitol at the γ-Al₂O₃(110)/water interface with the iso-surface of the average occupation volume of xylitol in ochre, and the average occupation volume of water in gray. The iso-surfaces were computed from an extra regular ab initio molecular dynamics simulation. More details can be found in Supplementary Note 8. **c** Free energy surface obtained from the metadynamics simulation on Al$_β$ in presence of xylitol. CN$_a$ is the coordination number of Al$_β$ to alumina oxygen atoms. CN$_{w+p}$ is the coordination number of Al$_β$ to water and polyols oxygens. Yellow and white balls represent aluminum and hydrogen atoms, respectively. The color red is used for alumina oxygen atoms and the associated CN$_a$. The color blue is used for water oxygen atoms and the associated CN$_w$

octahedral Al$_{(1)}$ and Al$_{(2)}$, which are in close vicinity to the water-sensitive tetrahedral Al$_β$, with xylitol. The choice of the adsorption geometry (Fig. 3b and other views in Supplementary Fig. 7) is globally in agreement with earlier studies on polyol adsorption on alumina (Supplementary Note 8)[29]. The reactivity of Al$_β$ is described using again the coordination number of Al$_β$ to alumina oxygen atoms (CN$_a$) as a variable. Since now, oxygen atoms are also found in xylitol, we considered, as a second variable, the coordination number to all other oxygen atoms that are external to alumina (oxygen atoms of water molecules and polyol) and referred to as CN$_{w+p}$. The resulting free energy landscape in the presence of xylitol (Fig. 3c) is profoundly modified compared to the uncoated case (Fig. 2c). Despite the exploration of a 200 kJ mol$^{-1}$ energy span, the sampled (CN$_a$,CN$_{w+p}$) space is reduced, meaning that Al$_β$ is more constrained near its initial geometry at (CN$_a$,CN$_{w+p}$) = (3,1). A detailed look at the free energy landscape shows nevertheless that Al$_β$ is able to react with some surrounding oxygen atoms (CN$_{w+p}$ goes up to 4). The exact mechanism is however very different from that described above. With xylitol adsorbed, CN$_a$ first decreases from 3 to 2 and CN$_{w+p}$ increases from 1 to 2, without any clear minimum in-between. This corresponds to a S$_N$2 mechanism with a strong S$_N$1 character where the Al$_β$ leaches first from alumina to the outer surface (CN$_a$ decreases) before gaining an Al–O bond with water or xylitol (CN$_{w+p}$ increases). As discussed above, the hydration of Al$_β$ follows an addition/elimination mechanism in the absence of xylitol. In other words, while a water molecule could easily reach the Al$_β$ and perform a nucleophilic addition in absence of xylitol, it has become more difficult in presence of xylitol. The nucleophile must therefore either be immobilized or have difficulties to approach Al$_β$. This is indeed confirmed by the inspection of the trajectories: Al$_β$ does not react with physisorbed water molecules but rather with the chemisorbed alcohol moieties of xylitol. This

frustrated extraction of Al$_β$ out of the alumina surface is highly activated and shows a barrier of 161 kJ mol$^{-1}$, consistent with the experimentally observed inhibiting power of xylitol for hydroxides formation.

To better understand why Al$_β$ reacts with xylitol rather than water, we have performed a 25 ps long regular ab initio molecular dynamics simulation of the interface with xylitol adsorbed and compared to our recently published study work on the γ-Al₂O₃(110)/water interface[33]. The volume visited by physisorbed water molecules around Al$_β$ is greatly impacted by the presence of xylitol as shown in Fig. 3a, b. This is further confirmed by the radial distribution functions of Al$_β$ with physisorbed water molecules: the probabilities of finding physisorbed water molecules in the second coordination shell are indeed reduced and shifted to larger distances (Supplementary Fig. 8). This clearly evidences an important steric hindrance induced by xylitol, which is at the origin of the inhibition.

Once Al$_β$ sees its coordination number to alumina oxygen atoms diminished by one unit, it directly coordinates to two alcohol groups because of the constrained adsorption mode of xylitol (CN$_{w+p}$ increases by two units). The resulting square-based pyramidal structure readily captures a water physisorbed molecule to gain a saturated octahedral geometry, reaching (CN$_a$, CN$_{w+p}$) = (2,4). Two other adsorption modes of xylitol, which substitute the chemisorbed water molecules involved in the decomposition mechanism, have been tested. They both turn out to be able to inhibit the early stages of the decomposition, proving that the inhibition shown here does not depend on the orientation of the tridentate adsorbate, in spite of mechanistic differences as discussed in Supplementary Note 9.

Noticeably, the inhibitor does not directly interact with the weak spot, but with a neighboring site, thereby modifying the interfacial structure of the liquid. It substitutes the water molecules that are

involved in the first hydration steps and prevents the necessary proton reshuffling to generate $AlO_xH_y$ and therefore provides a strong stabilization of the surface of alumina. This is attributed to the γ-pentanetriol backbone of xylitol, a structural feature shared with sorbitol (a known good inhibitor) but not with glycerol (a known weak inhibitor)[28]. Our study therefore opens the road to a rational design of the structure of coating agents able to prevent γ-$Al_2O_3$ from decomposing in water.

In conclusion, the combined experimental and theoretical study presented here provides an atomistic mechanistic picture of the initial steps of the decomposition of γ-$Al_2O_3$ in water. The weak spot is a tetrahedral surface Al of the (110) surface. It undergoes successive addition of water/scission of Al-O, yielding to $AlO_xH_y$. This process is accompanied by proton reshuffling. The chemisorption of a polyol on a neighbor site inhibits this hydrolysis by replacing the water molecules that initiate the process and by limiting the access of water to this weak spot. Noticeably, the weak spot of this oxide is not located at kink or edges but at the heart of a given facet. This understanding opens the road to further improvement of inhibitors. Since γ-$Al_2O_3$ is an iono-covalent oxide and water potentially reacts during the process and may dissociate into ($OH^-$, $H^+$), gaining such an atomistic understanding is a rather challenging task, especially regarding computational chemistry where the use of biased ab initio molecular dynamic is necessary. The present achievement thus constitutes an unprecedented milestone in the understanding of solid/liquid interface transformation. The approach that we propose here could be insightfully applied to various other relevant systems in electrochemistry, geochemistry or material science, where the reactivity of the solid with water and possibly other liquids plays a crucial role.

## Methods

**Alumina materials syntheses.** Alumina C (for Commercial) was obtained by calcination of commercial boehmite (Sasol PURAL SB3) at 600 °C for 4 h. Alumina F (for Fibers) was synthesized by precipitation of aluminum nitrate (Al $(NO_3)_3$·9$H_2O$, 0.1 mol $L^{-1}$) in an aqueous medium. The pH was adjusted to an initial value of 8 by addition of sodium hydroxide (NaOH, 1 mol $L^{-1}$) and the resulting suspension was aged at 95 °C for 1 week. The final pH value of the suspension was 4.5. The solid was recovered by centrifugation and washed three times with water. This synthesis was adapted from the method described by Chiche et al.[44] and Jolivet et al.[45] Alumina P (for Plates) and R (for Rods) were obtained by an hydrothermal treatment of alumina C in water (alumina P) or in acidified water with acetic acid (alumina R). Typically, 15 g of alumina C was dispersed in 100 mL of water or acidified water (pH = 2). The mixture was heated at 200 °C for 10 h under mechanical stirring and autogeneous pressure in a stainless steel autoclave. During this step, alumina was dissolved and boehmite precipitation occurred. Boehmite nanoparticles properties (size, morphology, texture, etc.) are depending on the experimental conditions (pH, temperature, concentrations…)[45,46]. In an acidic medium, boehmite precipitates as rod-like nanoparticles. In neutral medium, nanoparticles adopt a plate-like morphology. After cooling at room temperature (RT), the solid phase was recovered by centrifugation, dried at 100 °C overnight and calcinated at 600 °C for 4 h.

**Material characterization.** XRD analyses were performed on powders with a Bragg'Brentano diffractometer (PANalytical X'Pert PRO MDP) using Cu Kα radiation. Diffractograms were obtained from $2\theta = 4$–$74°$ with a step of 0.033° and 5 s per step. Textural properties of aluminas were determined by $N_2$ sorption studies at 77 K using a Micromeritics ASAP 2000 instrument. The BET method was applied to determine the specific surface area. Transmission electron microscopy (TEM) images were obtained on a JEOL 2010 $LaB_6$ microscope operating at 200 kV. A dispersion of the sample crushed in ethanol was deposited on standard holey carbon-covered copper TEM grids.

**Adsorption experiments.** Polyol adsorption isotherms were performed on three alumina materials exhibiting different morphologies in a 100 mL stainless steel autoclave equipped with a mechanical stirring rod (Top Industrie). Aqueous solutions of xylitol and sorbitol (0.5 g $L^{-1}$, 1 g $L^{-1}$, 2 g $L^{-1}$, 4 g $L^{-1}$, 6 g $L^{-1}$, and 8 g $L^{-1}$) prepared using commercial polyols (Sigma Aldrich) and deionized water. Typically, 2 g of alumina were dispersed in 50 mL of an aqueous solution of polyol. After 2 h at 200 °C under autogeneous pressure (14 bar) and vigorous stirring, a sample of the liquid phase was taken and the solid phase was recovered by

centrifugation. Final concentration was determined by high-performance liquid chromatography (HPLC) analysis using a Shimadzu Rezex RXM-Monosaccharide $Ca^{2+}$ 8% column connected to a differential refraction detector (Shimadzu RID10A). The amount of polyol adsorbed was calculated from the difference between initial and final concentrations.

**Atomistic model.** The model of γ-$Al_2O_3$(110)/water interface was taken from our previous study on the characterization of interfacial water in contact with γ-$Al_2O_3$.[33] This model was built using the surface model proposed by Digne et al.[47] (and further improved by Wischert et al.[48]) and has been used for about 15 years to rationalize efficiently experimental data.[19] The simulation cell consists of a 10 Å thick $2 \times 2$ unit cell of γ-$Al_2O_3$ (110), which is surmounted with a 20 Å thick water layer and another 10 Å thick layer of void to avoid spurious confinement effects[49].

**Molecular simulations.** All ab initio molecular dynamics simulations (AIMD) reported here were performed using density functional theory (DFT) with the Gaussian and Plane Wave combined approach as implemented in CP2K/Quickstep[50–53] using the same model and set of parameters as in our previous work on the γ-$Al_2O_3$(110)/water interface[33]. The electrons were treated using the exchange correlation PBE functional[54] with a Grimme D3[55] correction. Core electrons were described using the Goedecker-Teter-Hutter (GTH) pseudo-potentials[56–58] and the valence density was developed on a double-zeta DZVP basis set along with an auxiliary plane wave basis set with cutoff energy of 400 Ry.

During AIMD, nuclei were treated within the Born-Oppenheimer approximation with a time step of 0.5 fs for the integration of the equations of motion. The temperature of the simulation was maintained at 330 K using the Canonical Sampling through Velocity Rescaling (CSVR) thermostat coupled to the system with a time constant of 100 fs[59]. All the systems presented here were thermalized for at least 5 ps before production of about 30 ps for statistical analysis.

For the well-tempered ab initio metadynamics simulations[60], Gaussian hills of 0.04 width and 3.3 kJ $mol^{-1}$ initial height were added in a two-dimensional set of collective variables, described by coordination numbers (CN) as defined in Plumed 2.0[61]. For each collective variable, the chosen CN gives the average number on a set of oxygen atoms coordinated to the probed Al atom. We used two different sets. In the first one ($CN_a$), the oxygen atoms belong to the surface structure of dry alumina. The second one includes all the other oxygen atoms of the simulations ($CN_w$ or $CN_{w+p}$, with w for water oxygens and p for polyol oxygens, respectively). The error associated with this procedure is lower than 2 kJ $mol^{-1}$. Further details are given in Supplementary Notes 6–9.

## Data Availability

The source data underlying Fig. 1d–f are available in SI. Raw data were generated at the PSMN and GENCI large-scale facilities. Derived data supporting the findings of this study are available as supplementary materials: a snapshot of each ab initio MD is provided as well as the grid-based data of the Free Energy Surface that are plotted in Figs. 2c and 3c.

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

## Acknowledgements

We are grateful to the Center Blaise Pascal and the Pôle Scientifique de Modélisation Numérique at the École Normale Supérieure de Lyon for HPC resources. We also thank also the SYSPROD project and AXELERA Pôle de Compétitivité for financial support (PSMN Data Center). This work was granted access to the HPC resources of IDRIS under the allocation 2017-A0010800609 made by GENCI.

## Author contributions

S.P. and M.C. conceived and supervised the project. R.R. conducted the ab initio molecular dynamic simulations. C.P. computed the average occupation volume. G.E. conducted the experimental work that was supervised by T.A., B.M., C.A. and C.A. All the authors discussed the results and commented on the manuscript. R.R. and M.C. prepared the manuscript with inputs from C.P., G.E., T.A., B.M., and S.P.

## Additional information

**Competing interests:** The authors declare no competing interest.

