## [Peer Review File · Nature Communications]

Reviewers' comments:

Reviewer #1 (Remarks to the Author):

In their contribution the authors discuss the decomposition of gamma alumina on the (110) facet. The work combines ab initio metadynamics calculations with polyol adsorption experiments, with the aim to determine weak spots, namely points where hydrolysis should start.

From the simulations the Al-beta on the (110) facet is identified as the most reactive site. Hydration is sequential and follows an addition/elimination mechanism, where the hydration steps are accompanied by deprotonation of some of the coordinating water molecules.

Decomposition reaction is also performed in the presence of an inhibitor and shows, in this case a different mechanism with higher free energy barriers. The adsorption geometry of polyol on alumina is already known from previous studies.

One of the main conclusion is that the weak spot is not located at a kink or an edge.

The question is how can the authors exclude that Al sites at kinks or edges do not also represent weak points,

eventually with energetically more favorable dissolution pathways?

Have the authors investigated hydrolysis at kinks or edges?

Can the role of kink and edges be excluded from the experimental point of view?

On a more technical level. What is the accuracy of the used methods?

What are the error bars on the free energy profiles, e.g. in Fig. 2?

The same question is also pertinent to Figure 3.

Overall I find that is an interesting, but somewhat limited and specialised contribution, which could published in a more specialised journal.

Reviewer #2 (Remarks to the Author):

The manuscript deals with the study of Al₂O₃ structures from a combined experimental and theoretical approach. The subject is interesting and the work performed is state-of-the-art. However, I have a few concerns:

1) The title is completely misleading, the shape control is only seen in the experiments but the link to the calculations does not provide the control. The authors could do that by using a modified version of the Wulff construction. Otherwise, shape control needs to be removed from the title.

2) How is the pH accounted for? The description is not clear.

3) The definition of the coordination numbers is crucial to the discussion and thus needs to be part of the core manuscript. The equation in the methods does not present a correct qualitative description that is missing in the main text (i.e. from what distance on the contribution to the coordination is 0, for instance).

4) The authors claim in the conclusions that gamma-Al₂O₃ is covalent. I am sure that this is a misprint.

5) Introduction is the ? really wanted

- 6) Line 80 "reactive interface involving bonds with a covalent character" there is no need for covalent, as this kind of bond have a large contribution of ionicity.
- 7) The linear fittings in Figure 1 need to be reconsidered. If they are meant that the correlation does not exist it is quite misleading.
- 8) Filling the space between slabs results in water properties that are exceedingly rigid (Bellarosa ACS Central Sci.). The authors need to provide evidence that this is not their case.
- 9) The authors shall reconsider stating that the methodology is good, lines 178-182 and again 219.
- 10) The structure of the adsorbed alcohols on the surface is hardly identifiable. Although the structure is clearly dynamic it would be worth having an idea of the structure.

Reviewer #3 (Remarks to the Author):

The manuscript by Reocreaux, et al. is focused on developing a microscopic picture of the reactions that occur at the interface between alumina and water. Specifically, a series of ab initio metadynamics simulations and experimental characterization techniques are used to develop a detailed view of the initial decomposition reactions that are likely to occur when gamma-alumina comes in contact with water. In addition, a set of different inhibitors on the surface of alumina are examined that provide additional insight into the surface reactivity and opens up the possibility of rationalizing the design of new inhibitor compounds that have improved performance.

Although there are existing ab initio molecular dynamics simulations of the alumina-water interface, to the best of my knowledge this is the first investigation that uses the metadynamics approach to examine the surface chemistry in detail. The main conclusion from this work is that the (110) surface of gamma-alumina initially undergoes a series of reactions that involve the stepwise addition of water molecules, Al-O bond breaking, and a Grotthuss-like proton shuttling mechanism that ultimately leads to the hydration of individual Al ions. By examining the role of different surface inhibitors, such as xylitol, the role of the hydrogen bond network in the overall reaction mechanism is further explained.

I found the use of the metadynamics to be quite interesting for this system as it provides a rather detailed view into the various reaction pathways that are possible. The methods used are considered to be state-of-the-art for these large, complicated systems and I have not found any errors in what is being presented. Overall, I view this to be a significant advance in our understanding of the initial reaction mechanisms that are involved. That being said, I am somewhat skeptical about the authors claim of how broadly applicable the results from this investigation are to solid/liquid interface systems. In this regard, I do not believe that these findings will be of interest to the broader research community, which would argue for publication in a more specialized journal instead of Nature Communications.

In the following, we reply to the comments one by one. The comments are shown in blue, the modifications in red.

Please note that the abstract has been rewritten to follow the requirements of Nature Communication regarding the length.

Reviewers' comments:

Reviewer #1 (Remarks to the Author):

In their contribution the authors discuss the decomposition of gamma alumina on the (110) facet.

The work combines ab initio metadynamics calculations with polyol adsorption experiments, with the aim to determine weak spots, namely points where hydrolysis should start.

From the simulations the Al-beta on the (110) facet is identified as the most reactive site. Hydration is sequential and follows an addition/elimination mechanism, where the hydration steps are accompanied by deprotonation of some of the coordinating water molecules.

Decomposition reaction is also performed in the presence of an inhibitor and shows, in this case a different mechanism with higher free energy barriers.

The adsorption geometry of polyol on alumina is already known from previous studies.

One of the main conclusion is that the weak spot is not located at a kink or an edge.

The question is how can the authors exclude that Al sites at kinks or edges do not also represent weak points,

eventually with energetically more favorable dissolution pathways?

Have the authors investigated hydrolysis at kinks or edges?

Can the role of kink and edges be excluded from the experimental point of view?

Reply.

To start, we would like to thank the referee for the careful reading. He/She stresses here indeed a key point of our paper that was not enough underlined. As it is very often claimed and demonstrated, the kinks and edges can play a fundamental role in dissolution and are often identified as weak spots. In our study on the stability of gamma-alumina prepared for being a support in heterogeneous catalysis, the main exposed facet is the 110. It is identified as the location of the weak spots experimentally comparing the inhibiting coverage to the fractional area of a given facet of four sample of alumina with different morphologies. Following the same approach, those experiments allow us also to discard the kink and edges as a source of weak spots. This is now detailed in SI and acknowledged in the main text.

Modification in the main

We **cannot** establish such correlations with the other fractional surface areas. **Following the same kind of reasoning, we show in Supplementary Section B4 that edges and kinks are unlikely to be involved as major active sites for the hydrolysis.**

Modifications in SI:

1. Supplementary Table 2 has been completed with the amount of edges in each sample:

Sizes and shapes of the nanoparticles in alumina P, C and F (resp. plates, commercial and fibres) as determined from XRD analysis (**Supplementary Table 2**) are in good agreement with the TEM observations (**Fig. 1g-j**). However, this is not the case for alumina R (rods). According to TEM pictures, this alumina should present a very high proportion of lateral (100) facets (*i.e.* dimension $y \gg x$ and e) and a small proportion of (111) facets. This difference indicates that boehmite R platelets at the origin of alumina R are most likely polycrystalline. The polycrystallinity of such boehmite nanorods has already been observed by Mathieu *et al.*⁶. More generally, the morphologies obtained in the current work are in good agreement with the recent work by Lee *et al.*⁵ that determined the morphologies of boehmite platelets using TEM images.

Supplementary Table 2 | Characterisation of the morphology of the four samples of $\gamma\text{-Al}_2\text{O}_3$ consisting of nanoparticles with different edge lengths and facet surface area distributions. S_{BET} is the specific surface area. F, R, P and C respectively stand for fibres, rods, plates and commercial alumina.

	S_{BET} (m ² /g)	$e_{\text{Al}_2\text{O}_3}$ (nm)	$x_{\text{Al}_2\text{O}_3}$ (nm)	$y_{\text{Al}_2\text{O}_3}$ (nm)	(110) facet (%)	(100) facet (%)	(111) facet (%)
Alumine F	250	1.9	2.5	5.7	60	21	19
Alumine R	179	4.8	8.4	11.5	64	15	22
Alumine P	78	16.9	21.4	6.4	50	7	43
Alumine C	212	2.2	6.6	3.7	73	9	18

2. A new table has been added as **Supplementary Table 5** to give the amount of polyol adsorbed on the different samples of alumina in nm⁻¹. It comes with a description of how such “lineal coverages” have been estimated.

Under the assumption that both sorbitol and xylitol adsorb specifically to edges, one can also define a lineal inhibiting coverage. To do so, we can first determine the total length D of edges for one particle in each sample using the dimensions given in **Supplementary Table 2** (see equation (iv)). From these dimensions, one can also determine the surface area of one particle S_{particle} . The amount N of particles in each sample can then be estimated from the BET specific surface area, the mass of the alumina sample m and S_{particle} (see equation (v)). Dividing the amount n_{ads} of polyol adsorbed by the product $N \times D$ we obtain the lineal coverage.

$$D = 8x + 6e + 4y \quad (\text{iv})$$

$$N = \frac{S_{\text{BET}} \times m}{S_{\text{particle}}} \quad (\text{v})$$

$$\Lambda = \frac{n_{\text{ads}}}{N \times D} \quad (\text{vi})$$

Supplementary Table 5 | Lineal inhibiting coverage for sorbitol and xylitol for each sample of alumina.

Alumina	Sorbitol inhibiting coverage (sorbitol.nm ⁻¹)	Xylitol inhibiting coverage (xylitol.nm ⁻¹)
C	1.06 ± 0.08	0.90 ± 0.12
P	2.76 ± 0.89	1.92 ± 1.30
R	1.41 ± 0.15	1.18 ± 0.25
F	0.51 ± 0.04	0.42 ± 0.06

3. A section (Supplementary Section B4) has also been included to show that we can rule out the involvement of kinks and edges on the basis of the characterisation of the shape controlled nanoparticles. It comes with a figure (Supplementary Fig. 4) that shows that the amount of polyol adsorbed (which prevent the decomposition of alumina) does not correlate with the amount of edges.

B4. Can kinks and edges be involved in the decomposition mechanism of alumina?

First, the amount of kinks is much smaller than the amount of polyol required to inhibit the decomposition of alumina. It is therefore very unlikely that they play a major role in the decomposition mechanism of alumina. As for the edges, the situation is a bit more delicate. In a polydentate configuration, the size of either xylitol or sorbitol is about 1 nm, meaning that lineal coverages cannot go beyond 1 nm⁻¹. This corresponds to the order of magnitude of lineal inhibiting coverages determined experimentally and given in **Supplementary Table 5**. The fact that lineal coverages can experimentally exceed 1 nm⁻¹ is already a bit suspicious but could be explained with a certain amount of monodentate species at the edges. We thus need to analyse edges in more details.

To understand the potential role of edges, we have followed the same analysis as that carried out for the facets and described in **Fig. 1**. We assume in this paragraph that the weak spots are on the edges, i.e. decomposition is initiated at the edges and polyols preferably interacts with the edges thereby providing protection of the nanoparticles.

First, it is worth noting that the lineal inhibiting coverages vary between 0.51 and 2.76 nm⁻¹ and 0.42 and 1.92 nm⁻¹ for sorbitol and xylitol respectively. From one sample to another, only the shape of the nanoparticles changes: the three types of edges (namely E, X and Y see **Supplementary Fig. 4**) are present in different proportions (see **Supplementary Table 2**).

The strong variation of the lineal inhibiting coverages (up to a factor of 5) therefore suggests that adsorption might occur more preferably at one specific edge. When we plot the lineal inhibiting coverage as a function of the fractional length, we see however no correlations ($R^2 < 0.9$). This means that there are no such things as specific interaction with one edge in particular.

The shape effect evidenced by the strong variation of (lineal) inhibiting coverage can seemingly be solely explained for polyol molecules interacting specifically with the (110) facet, as shown in **Fig. 1**.

This is consistent with previous work by Copeland *et al.*⁸ and Larmier *et al.*^{9,10} who were able to explain the properties of moderately hydrated alumina regarding the spectroscopy and

reactivity of alcohols/polyols without invoking edges and kinks. It is very likely that the Lewis acid and basic sites at edges and kinks are almost instantaneously saturated with water at the early stage of water adsorption. The resulting aluminol groups are most probably strongly bound to the edges/kinks and cannot be displaced with alcohol/polyol molecules.

Supplementary Figure 4 | Non-implication of edges in the inhibiting adsorption of polyols on the four alumina samples. **a**, General topology of a γ - Al_2O_3 nanoparticle exhibiting three major edges: E in black, X in purple and Y in green. **b,c,d**, Comparisons between the *lineal inhibiting coverage* (sorbitol in red, xylitol in blue) and the fraction of exposed E (**d**), X (**e**) and Y (**f**) edge lengths. The experimental data sets are fitted to the zero-intercept linear model derived in **Supplementary Section B3**. The fitted curves are represented as straight lines. No correlation could be established for any edges ($R^2 < 0.9$).

On a more technical level. What is the accuracy of the used methods?

What are the error bars on the free energy profiles, e.g. in Fig. 2?

The same question is also pertinent to Figure 3.

Reply:

The energies and forces are computed at the PBE+D3 level of theory, which is the best we can afford for such an extensive ab initio molecular dynamics. It comes with its own error, generally up to 20-30 kJ/mol on reaction energies and barriers. Lagauche et al, using similar method (PBE+D2) found a 14 kJ/mol shift between the experimental and theoretical maximum value of the energy distribution function of the water adsorption enthalpy on gamma-alumina, hence indicating that water adsorption strength is reasonably estimated by this level of theory. Then, the well-tempered metadynamics allows controlling the error and keeping it as low as possible. The size of the Gaussian bias (hills) is reduced with the exploration time of a given zone. In our case, it goes below $2\text{kJ}\cdot\text{mol}^{-1}$.

Modification in the main. A sentence has been added in the Method part to provide the error associated with the metadynamics.

The error associated with this procedure is lower than $2\text{kJ}\cdot\text{mol}^{-1}$.

Overall I find that is an interesting, but somewhat limited and specialised contribution, which could published in a more specialised journal.

Reply: gamma-Alumina is an essential support in Catalysis and understanding the mechanisms underlying its absence of stability in aqueous environment is a key challenge in the field. Beyond its significance in this important field, the interfacial properties of gamma-

alumina with water have also a broad interest for adhesion and other surface engineering domains. Indeed, gamma-alumina can also be exposed on aluminium as a protective oxidative layer. Last, solid/liquid interfaces are at the heart of a wide range of domains (from geochemistry to pharmaceutical sciences) and in particular the stability/dissolution of a solid in contact with a liquid. It raises more and more interest thanks to progresses in directed synthesis and computational chemistry as we show here but also in operando spectroscopies (SFG, SHG, DNP, etc.). Thus, this frontier becomes more and more accessible and we believe that the approach that we propose here combining computational chemistry and reactivity of shaped-control particles could be transferred to interfaces of interests in other fields.

Modification in the main

None.

Reviewer #2 (Remarks to the Author):

The manuscript deals with the study of Al₂O₃ structures from a combined experimental and theoretical approach. The subject is interesting and the work performed is state-of-the-art.

Reply. We would like to thank the reviewer for the positive appreciation of our work and the constructive remarks to improve the manuscript.

However, I have a few concerns:

1) The title is completely misleading, the shape control is only seen in the experiments but the link to the calculations does not provide the control. The authors could do that by using a modified version of the Wulff construction. Otherwise, shape control needs to be removed from the title.

Reply: We agree that the title was confusing. The shape-control was related to the synthesis of alumina particles only, not the simulations. We have now modified it to take into account this remark and clarify the title.

Modification to the main:

The title has been changed to:

Reactivity of shape-controlled crystals and metadynamics simulations locate the weak spots of alumina in water

2) How is the pH accounted for? The description is not clear.

Reply: Experimentally, the pH is the one of deionized water in contact with gamma-alumina whose iso-electric point is at ~7-8, i.e. the pH of the solution is around 7. In our simulation, we used bulk neutral water, which matches this experimental pH.

Modification to the main:

A sentence has been added in the Results and discussion part page 5:

... Repeating the surface unit cell in the x and y directions, the resulting p(2x2) slab is surmounted by a 20 Å thick layer of liquid water that is not represented here for clarity. The iso-electric point of alumina being close to 8, we consider pure neutral water^{41,42}. Since the decomposition of γ -Al₂O₃ transforms tetrahedral Al sites into octahedral Al centers,...

3) The definition of the coordination numbers is crucial to the discussion and thus needs to be part of the core manuscript. The equation in the methods does not present a correct qualitative description that is missing in the main text (i.e. from what distance on the contribution to the coordination is 0, for instance).

Reply: We fully agree that the definition of the coordination numbers is crucial to the discussion and needs an improved description. We modified extensively the corresponding description in the main text. We added a paragraph to the **Supplementary Section C** with a description of the function we used as a coordination number and a plot of one of its component along with a paragraph to explain the influence of each parameter.

Modification to the main (Methods section)

For the well-tempered *ab initio* metadynamics simulations⁵⁷, Gaussian hills of 0.04 width and 3.3 kJ.mol⁻¹ initial height were added in a two-dimensional set of collective variables, described by coordination numbers (CN) as defined in Plumed 2.0⁵⁸. For each collective variable, the chosen CN gives the average number on a set of oxygen atoms coordinated to the probed Al atom. We used two different sets. In the first one (CN_a), the oxygen atoms belong to the surface structure of dry alumina. The second one includes all the other oxygen atoms of the simulations (CN_w or CN_{w+p}, with w for water oxygens and p for polyol oxygens respectively). Further details are given in **Supplementary Section C**.

Modification to the SI

The collective variables we chosen are based on coordination numbers. To describe the coordination of aluminium Al_j with a set of oxygens {O_{set}} CN is defined by:

$$CN(Al_j, O_{set}) = \sum_{i \in \{O_{set}\}} s_{ij}(r_{ij}) = \frac{1 - \left(\frac{r_{ij} - d_0}{r_0}\right)^n}{1 - \left(\frac{r_{ij} - d_0}{r_0}\right)^m}$$

with r_{ij} the inter-atomic distance between atom i and atom j , $s_{ij}(r_{ij})$ the switching function describing the coordination between atom i and j , d_0 the central value of the switching function ($s_{ij}(d_0) = 1$), r_0 the acceptance distance of the switching function, and with n and m two integer exponents with $n < m$. The switching function s_{ij} is plotted for a given value of r_0 and various ratios n/m to illustrate the effect of those parameters on its shape in **Supplementary Figure 4**. d_0 is chosen to match the Al-O equilibrium distance (1.5Å). r_0 can be seen as an acceptance distance which, coupled with the n/m ratio, controls at which distance the O atom is not anymore considered as bonded to Al. At d_0+r_0 , the switching function s_{ij} is equal to n/m . The n/m ratio can be seen as the swiftness of decrease of the function away from the equilibrium distance d_0 . Here, a ratio of 2/5 ($n=4$, $m=10$) and a r_0 of 0.9Å has been chosen and we can, therefore, observe that this switching function has a value of about zero for an interatomic distance greater than 3Å.

Supplementary Figure 4 | Switching function s_{ij} for $r_0 = 0.9 \text{ \AA}$ and n/m ratio with $n = 4$. The corresponding mathematical definition is given in the text.

4) The authors claim in the conclusions that gamma-Al₂O₃ is covalent. I am sure that this is a misprint.

Reply: We agree that this was a misprint; gamma-alumina is an **iono-covalent** oxide (see Digne et al. 2004 for instance). We wished to underline that this is not a purely ionic solid that could have been described more easily using a force field description. Using an ab initio approach is mandatory on this system while it would not be indispensable on NaCl for instance (see for instance Aragones et al. *J. Chem. Phys.* (2012) 136, 244508). We have corrected that through the text.

Modifications to the main:

Ab initio metadynamics can hence provide a complete atomistic picture of the alteration mechanism of an iono-covalent solid in contact with a reactive solvent.

...

This understanding opens the road to further improvement of inhibitors. Since γ -Al₂O₃ is an iono-covalent oxide and water potentially reacts during the process and may dissociate into (OH⁻, H⁺), gaining such an atomistic understanding is a rather challenging task, especially regarding computational chemistry where the use of biased *ab initio* molecular dynamic is necessary.

5) Introduction is the ? really wanted

Reply: no, it is not.

Modification to the main:

It has been removed.

6) Line 80 "reactive interface involving bonds with a covalent character" there is no need for

covalent, as this kind of bond have a large contribution of ionicity.

Reply: See point 4. There is a need of covalent since this implies to use an ab initio approach and not a force field. We have rephrased the incriminated sentence to better underline that point.

Modification to the main:

Although common using classical force field for crystal growth and dissolution of molecular and ionic crystals^{39,40}, *ab-initio* metadynamics has never been used to describe a reactive interface involving bonds with a **partial** covalent character (typically here Al-O and O-H bonds) **hence requiring an *ab initio* description.**

7) The linear fittings in Figure 1 need to be reconsidered. If they are meant that the correlation does not exist it is quite misleading.

Reply: We have revised our representation of the zero-intercept linear fits and the caption to avoid any misleading interpretation of the straight lines. The caption now properly refers to the Supplementary Information, where the linear model is derived. A comment on what is considered a good fit has also been added in the caption.

Modification to the main:

The Figure 1 has been modified accordingly.

Figure 1 | Implication of the (110) facet in the decomposition of γ -Al₂O₃. a, General topology of a γ -Al₂O₃ nanoparticle exhibiting three major facets: (110) in dark blue, (100) in light blue and (111) in grey. b,c, Structures of two inhibitors: b, sorbitol (red) and c, xylitol (blue). d,e,f, Comparisons between the *inhibiting coverage* (sorbitol in red, xylitol in blue) and the fraction of exposed (110) (d), (111) (e) and (100) (f) surface areas. **The experimental data sets are fitted to the zero-intercept linear model derived in Supplementary Section B3. The fitted curves are represented as straight lines.** A correlation could only be established in the case of the (110) surface (d) with slopes of 0.367 ± 0.009 and 0.443 ± 0.005 nm² for

xylitol and sorbitol respectively ($R^2 > 0.99$). **g,h,i,j**, Transmission Electronic Microscopy images of the four differently shaped γ -Al₂O₃ nanoparticles ordered in increasing (110) fractional area: plates (**g**), fibres (**h**), rods (**i**) and commercial γ -Al₂O₃ (**j**).

8) Filling the space between slabs results in water properties that are exceedingly rigid (Bellarosa ACS Central Sci.). The authors need to provide evidence that this is not their case.

Reply: We fully agree that filling the space between the slabs would likely result in water properties that are not adequate (confined water). We have here chosen a non-symmetric approach where the alumina slab is surmounted by a water slab and then separated from the next image of the slab by a vacuum of 10 Å. More details can be found in our previous paper (Réocreux et al. ACS Appl. Nano Mater. 1, 191–199 (2018)) and in the **Methods/Atomistic model** section: “The simulation cell consists of a 10 Å thick 2x2 unit cell of γ -Al₂O₃ (110), which is surmounted with a 20 Å thick water layer and another 10 Å thick layer of void.” We underlined this choice citing the paper of Bellarosa et al.

Modification to the main:

The simulation cell consists of a 10 Å thick 2x2 unit cell of γ -Al₂O₃ (110), which is surmounted with a 20 Å thick water layer and another 10 Å thick layer of void to avoid spurious confinement effects⁴⁷.

9) The authors shall reconsider stating that the methodology is good, lines 178-182 and again 219.

Reply: Our aim was to underline that the methodology could be transferred to other solid/liquid interfaces. We have rephrased the first sentence and removed the second one.

Modification to the main:

“It also demonstrates how powerful ab initio metadynamics can be at providing a complete atomistic picture of the alteration mechanism of a covalent solid in contact with a reactive solvent.”

becomes

“*ab initio* metadynamics can hence provide a complete atomistic picture of the alteration mechanism of a iono-covalent solid in contact with a reactive solvent.”

“Here again, the robustness of ab initio metadynamics at inspecting reactive interface is confirmed.”

has been removed.

10) The structure of the adsorbed alcohols on the surface is hardly identifiable. Although the structure is clearly dynamic it would be worth having an idea of the structure.

Reply: We agree that the figure does not provide an easily identifiable view of the adsorption of the alcohol. The aim was to shed light on the restructuring of water at the interface in presence of the alcohol. We have added a set of other views of a snapshot of the polyol at the interface in SI in section **C3.1 Sites of adsorption** and added a reference to them in the main. We also added the .xyz file of this snapshot so that anyone can visualize it easily and added a sentence in the Data Availability section.

Modification to the main:

The choice of the adsorption geometry (**Fig. 3b** and other views in **Supplementary Fig. 6**) is globally in agreement with earlier studies on polyol adsorption on alumina (**Supplementary Section C3.1**)²⁹.

Data Availability

The source data underlying Fig. 1d, 1e, 1f are available in SI. A snapshot of each ab initio MD is provided as SI as well as the grid-based data of the Free Energy Surface that are plotted in Fig. 2c and Fig. 3c.

Modification to the SI:

The supplementary Figure 6 has been complemented with:

and its caption has been modified accordingly.

Supplementary Figure 7 | Adsorption of glycerol and xylitol through the substitution of water molecules. **a**, Structure of γ - $\text{Al}_2\text{O}_3(110)$ surface saturated with water molecules. **b**, Substitution of chemisorbed water molecules with glycerol (structure proposed by Copeland *et al.*⁸) and xylitol (geometry proposed in the present work) and various views of a snapshot of xylitol at the Al_2O_3 -water interface (O in red except O from water in blue, Al in yellow, C in black and H in white).

Reviewer #3 (Remarks to the Author):

The manuscript by Reocreux, et al. is focused on developing a microscopic picture of the reactions that occur at the interface between alumina and water. Specifically, a series of ab

ab initio metadynamics simulations and experimental characterization techniques are used to develop a detailed view of the initial decomposition reactions that are likely to occur when gamma-alumina comes in contact with water. In addition, a set of different inhibitors on the surface of alumina are examined that provide additional insight into the surface reactivity and opens up the possibility of rationalizing the design of new inhibitor compounds that have improved performance.

Although there are existing ab initio molecular dynamics simulations of the alumina-water interface, to the best of my knowledge this is the first investigation that uses the metadynamics approach to examine the surface chemistry in detail. The main conclusion from this work is that the (110) surface of gamma-alumina initially undergoes a series of reactions that involve the stepwise addition of water molecules, Al-O bond breaking, and a Grotthuss-like proton shuttling mechanism that ultimately leads to the hydration of individual Al ions. By examining the role of different surface inhibitors, such as xylitol, the role of the hydrogen bond network in the overall reaction mechanism is further explained.

I found the use of the metadynamics to be quite interesting for this system as it provides a rather detailed view into the various reaction pathways that are possible. The methods used are considered to be state-of-the-art for these large, complicated systems and I have not found any errors in what is being presented. Overall, I view this to be a significant advance in our understanding of the initial reaction mechanisms that are involved. That being said, I am somewhat skeptical about the authors claim of how broadly applicable the results from this investigation are to solid/liquid interface systems. In this regard, I do not believe that these findings will be of interest to the broader research community, which would argue for publication in a more specialized journal instead of Nature Communications.

Reply:

We could like to thank the reviewer for his/her careful reading. As underlined here, this study is likely to be the first one to use a rare event approach (metadynamics here) to investigate in details surface chemistry. We are now able to provide a complete picture of elementary steps involving bond breaking and bond formation at the interface between water and an ionic-covalent solid. This approach can be clearly of a wide interest beyond the application fields of gamma-alumina in contact with water (namely Catalysis but also surface engineering such as adhesion, etc) and inspire works in geochemistry, pharmaceutical sciences, etc. as emphasized in the Introduction.

REVIEWERS' COMMENTS:

Reviewer #1 (Remarks to the Author):

I think that the authors have addressed the points raised by the reviewers and the manuscript has improved in the revision process. I therefore recommend the publication.

Reviewer #2 (Remarks to the Author):

The authors have addressed all the points I raised in previous correspondence. I particularly appreciate that they have reflected on the title issue to incorporate both experiments and theory. Therefore, I can recommend the manuscript for publication in its present form.